# A Foggy Weather Simulation Algorithm for Traffic Image Synthesis Based on Monocular Depth Estimation

**DOI:** 10.3390/s24061966

**Published:** 2024-03-20

**Authors:** Minan Tang, Zixin Zhao, Jiandong Qiu

**Affiliations:** 1College of Automation and Electrical Engineering, Lanzhou Jiaotong University, Lanzhou 730050, China; 2College of Electrical and Mechanical Engineering, Lanzhou Jiaotong University, Lanzhou 730070, China; 12221501@stu.lzjtu.edu.cn (Z.Z.); qiujd@mail.lzjtu.cn (J.Q.)

**Keywords:** traffic safety, monocular depth estimation, atmospheric scattering model, fog simulation, natural scene statistics

## Abstract

This study addresses the ongoing challenge for learning-based methods to achieve accurate object detection in foggy conditions. In response to the scarcity of foggy traffic image datasets, we propose a foggy weather simulation algorithm based on monocular depth estimation. The algorithm involves a multi-step process: a self-supervised monocular depth estimation network generates a relative depth map and then applies dense geometric constraints for scale recovery to derive an absolute depth map. Subsequently, the visibility of the simulated image is defined to generate a transmittance map. The dark channel map is then used to distinguish sky regions and estimate atmospheric light values. Finally, the atmospheric scattering model is used to generate fog simulation images under specified visibility conditions. Experimental results show that more than 90% of fog images have AuthESI values of less than 2, which indicates that their non-structural similarity (NSS) characteristics are very close to those of natural fog. The proposed fog simulation method is able to convert clear images in natural environments, providing a solution to the problem of lack of foggy image datasets and incomplete visibility data.

## 1. Introduction

The continuous development of artificial intelligence technology has pushed self-driving cars to the forefront of societal discussion, raising significant concerns about their safety. According to data collected by the World Health Organisation (WHO) from 178 countries up to 2023, 1.19 million people worldwide lose their lives each year due to road traffic accidents [1]. Road traffic injuries have huge economic costs for individuals, families, and entire countries, including the cost of treatment for the dead and injured as well as the loss of labour for the deceased, those disabled by their injuries, and family members who need to take time away from work or school to care for the injured. Statistically, losses from road traffic accidents account for 3 percent of the gross domestic product of most countries. The main causes of road accidents around the world include human factors, road factors, vehicle factors, and environmental factors. Among these factors, adverse weather conditions play a crucial role. The focus on adverse weather conditions is of great significance in research directed towards road traffic safety. The data and background information highlight the potential benefits of self-driving cars, especially in terms of improving road traffic safety. As artificial intelligence technology continues to advance, self-driving cars can adapt to adverse weather conditions by sensing and responding in real time, reducing the number of accidents. The application of such technology is expected to reduce the number of road deaths globally and alleviate the heavy burden borne by individuals, families, and national economies. Overall, it is more realistic to strengthen the research and application of autonomous driving technology, especially in bad weather conditions, while concentrating on foggy target detection, which is more challenging due to the problems of colour distortion and the loss of detailed features in foggy images, which bring many difficulties in extracting foggy image features. This will not only help reduce casualties and property damage but also promote sustainable socio-economic development.

In recent years, researchers in the field of target detection have shifted from traditional image processing methods to convolutional neural network (CNN)-based approaches [2,3,4,5,6,7,8] focusing on image defogging algorithms that improve scene visibility. This shift has triggered a significant amount of research aimed at improving image quality and enhancing the performance of target detection. Codruta et al. [7] proposed a new single-image defogging method for improving the visibility of foggy images. The method of the paper is a fusion-based strategy that extracts inputs from two original blurred image by applying white balance and contrast enhancement procedures. Filtering of luminance, chromaticity, and saliency features by calculating three metrics is used to efficiently blend the derived inputs in order to retain regions with good visibility; the multi-scale approach is designed using a Laplace pyramid representation to minimise artefacts introduced by the weight map. Liu et al. [9] proposed to add a dark channel fog removal algorithm based on the YOLOv7 algorithm and effectively improve the fog removal efficiency of the dark channel through the methods of downsampling and upsampling.

Although many algorithms have achieved the goal of removing haze, the introduction of defogging algorithms into foggy detection channels still faces high costs in practical applications. This means that deploying the algorithms on resource-constrained devices or applications that need to run in real time is expensive and infeasible. Therefore, it is particularly important to improve the accuracy of foggy road traffic target detection models without adding extra costs. For data-driven algorithms, the amount and diversity of training data directly affects the performance of the model. However, the lack of foggy weather datasets becomes a major obstacle to improve the performance of foggy weather target detection models. Therefore, in order to improve the accuracy of target detection in foggy weather conditions, the size of the dataset containing road traffic scenes must be significantly increased.

According to the above study, in order to scale up a traffic scene dataset, the dataset must meet certain criteria: the dataset should cover a wide range of application scenarios, show a variety of categories, contain a variety of environmental features, and preferably have a high frame rate. Well-known open-source annotated datasets commonly used for target detection in autonomous driving traffic scenarios include KITTI [10], Cityscapes [11], and BDD100K [12]. After careful evaluation, the BDD100K dataset emerged as the most suitable choice, as it contains a wider variety of data categories than similar datasets. In addition, the dataset is geographically, environmentally, and meteorologically diverse, which greatly facilitates the development of data-driven models. Extending the dataset to include foggy conditions is a feasible approach, where foggy traffic images are artificially synthesised from sunny day images. This approach allows trained models to improve performance in all scenarios. Similar approaches have been used to augment existing datasets for scenes such as rainy [13] days and nighttime [14]. For example, the literature [15] discusses the use of synthetic images for lane line detection, while previous studies have investigated synthetic fog images for automated driving, including FRIDA [16], FRIDA2 [17], Foggy Cityscape [18], and Multifog KITTI [19]; few studies have quantitatively evaluated artificially generated fog image enhancement datasets to determine their fidelity. Tarel et al. constructed digital image-based synthetic outdoor fog datasets, namely FRIDA (Foggy Road Image Database) and FRIDA2, using $SiVIC^{TM}$ software, but these datasets contain a limited number of images with low resolution and poor visual quality and are therefore insufficient to support the training of target detection models with higher accuracy and on a larger scale, especially in fog scenes. Sakaridis et al. developed Foggy Cityscape, and Mai et al. introduced synthetic fog into Cityscapes and KITTI images to create Multifog KITTI, but the scale of these datasets remains relatively small. In addition, these fog image synthesis methods rely on image perception and depth information acquired by sophisticated sensors such as LiDAR, RGBD cameras, or binocular cameras, which adds additional cost in practice.

In this paper, we propose a new dataset enhancement method to extract depth information from monocular clear weather images using a self-supervised monocular depth estimation method, and then simulate foggy weather images based on an atmospheric scattering model. Our proposed framework estimates the depth map only from the original image, which means that the method can be easily applied to other datasets without the need for additional sensors to collect depth information. To improve the accuracy of the target detection model in foggy scenes, we simulate foggy images using the method in this paper on the BDD100K dataset, resulting in the FoggyBDD100K dataset. The quality assessment of our simulated FoggyBDD100K dataset verifies the authenticity of our obtained dataset and the effectiveness of this foggy sky simulation framework. The main contributions of this thesis include:(1)A new enhancement and simulation method for a foggy sky target detection dataset is proposed, which significantly improves the accuracy and generalisation of the foggy sky target detection model without introducing additional computational cost and complex algorithmic framework.(2)We have created a new dataset, FoggyBDD100K, which contains 16,702 foggy sky images. This will help the community and researchers to develop and validate their own data-driven target detection or defogging algorithms.(3)We quantitatively evaluated the obtained simulated foggy sky images and obtained quantitative evaluation results while validating the reliability of this foggy sky simulation framework.

Given the above research, traffic safety becomes a necessary factor in the study of severe weather target detection. About the rest of this article: Section 2 describes the methods used in this paper. Section 3 comparatively analyses and presents the reference-free image quality assessment algorithm used in this paper for fog map assessment. Section 4 describes the experimental procedure and analyses the experimental results. Finally, Section 5 summarises the work of this paper.

## 2. Methods

This study presents a foggy sky simulation algorithm developed for the BDDK100 dataset. The method uses a self-supervised monocular depth estimation method to extract relative depth information from monocular images. Next, a dense geometric constraint method is used to determine the absolute depth information in the image. A simulated haze image is then generated using an atmospheric scattering model. Finally, the simulated haze image is quantitatively assessed using the AuthESI image quality assessment algorithm.

### 2.1. Self-Supervised Relative Depth Estimation Network

The acquisition of accurate scale information directly from monocular images poses a common challenge for researchers, often necessitating the use of lidar [20,21,22] or stereo cameras [23,24,25,26,27] for depth estimation. Among stereo methodologies, Mo [24] proposed a fusion method that combines the advantages of tightly coupled depth sensors and stereo cameras to complete dense depth estimation, thereby achieving better depth estimation. Xiaofeng [27], on the other hand, studied accurate depth estimation under different mechanical parameters such as camera calibration and alignment errors. ManyDepth [28] leverages the matching geometric properties across time-series images to enhance the accuracy of single-image depth prediction. However, under these related methods, the cost increases significantly, which undoubtedly indirectly increases the difficulty of generalisation. In addition, the bulky structure reduces the speed of reasoning, which in turn reduces the applicability of the methods.

The focus of this study is to achieve accurate relative depth estimation using the Monodepth2 benchmark network, which employs a self-supervised monocular depth estimation network. The algorithmic framework consists of two main components: the depth estimation network and the pose estimation network. The task of the depth estimation network is to estimate the depth value of each input image at the pixel level. By utilising the depth estimation function of Monodepth2, this network can accurately compute the depth map, thus enhancing the overall depth estimation process. Meanwhile, the pose estimation network is responsible for estimating the six-degree-of-freedom poses between neighbouring input image frames. Its main function is to construct self-supervised signals between images through geometric transformations. This network of the algorithm is crucial for building robust self-supervised learning signals that help the network to refine the depth estimation through geometric relationships in the image sequence.

This section delves into the intricacies of the unsupervised monocular depth estimation framework, with a focus on the depth estimation network responsible for obtaining depth information about an image. This network is the cornerstone of the research and the core of the article. The depth estimation network is implemented through a U-Net-based encoder–decoder network architecture, the architecture of which is crucial for understanding and optimising monocular depth estimation methods. The encoder employs multiple convolutional layers, each of which performs downsampling operations to generate feature maps at different scales. As shown in Figure 1, the Monodepth2 Depth Estimation Network Architecture Diagram visually summarises the intricate architecture and interrelationships between components in the depth estimation network. This analysis provides a comprehensive understanding of the network functionality and lays the foundation for advancing unsupervised monocular depth estimation methods.

In order to solve the challenges posed by bilinear sampling and local gradients caused by local minima in the current study, the baseline network DNet in the literature [29] was incorporated. The commonality with reference [30,31,32,33] is the use of a multi-scale depth prediction strategy. To further mitigate depth artefacts and enhance object-level depth inference, a novel Disparity Correction and Propagation (DCP) layer is introduced. The proposed DCP layer explicitly merges features at different scales in a hierarchical manner. The design rationale for the layered structure stems from the observation that low-resolution layers within the decoder network provide more reliable object-level depth inference, while high-resolution layers focus on capturing local depth details. The algorithm uses DCP to initiate the depth prediction process, perform relative depth estimation, and generate structural features.

Specifically, a convolutional neural network in the DCP layer is used to normalize the number of feature channels at different scales to eight, which is used to simplify the subsequent computation. Subsequently, features from the low-resolution layer are upsampled and feature fusion is performed with features from the high-resolution layer. This fusion introduces more accurate object-level inference to the higher-resolution depth prediction, thus improving the accuracy of the generated depth maps.

### 2.2. Absolute Depth Scale Recovery Based on Monodepth2

This module utilizes the geometric relationship between the ground and the camera in the dataset, focusing on extracting suitable ground points and then calculating the distance from the camera to the actual pixels in the image. It works by selecting ground points with stability and scale consistency as landmarks for the initial scale calculation. This assumption is in line with the general case of automated driving, where a sufficient number of ground points in a monocular image can be recognized.

After obtaining the relative depth map through the relative depth network, the module starts the scale recovery. The relative depth values are the basis for projecting the coordinates of each pixel point in the image into 3D space, and they follow the principles outlined in Equation (Equation 1). The result of this projection forms the corresponding 3D coordinates, paving the way for a robust ground-based scale recovery process.
(1)$$D_{t}^{rel}\left(p_{i,j}\right)p_{i,j}=KP_{i,j},$$
where $p_{i,j}$ represents the pixel point in the *i*-th row and *j*-th column with homogeneous coordinates in two-dimensional space, $P_{i,j}$ is a 3D point in the corresponding 3D space, $D_{t}^{rel}\left(p_{i,j}\right)$ is the relative depth corresponding to a pixel point in the 2D space, and *K* is the camera internal matrix.

Subsequently, in this paper, the eight-point method is used to calculate the surface normal vector. The eight-point method is a method to determine the surface normal vector of a pixel point by considering eight neighboring pixel points around the pixel. As shown in Figure 2, the red point in the center is the pixel point for which the surface normal vector is to be determined, and the eight pixel points around it are divided into four groups according to a specific rule, with each group containing two points. According to Equation (Equation 2), two sets of vectors are constructed starting from the red pixel point and ending at two pixel points in each group, ensuring that each group of points is perpendicular to the vector of the center pixel point, thus creating planes that remain orthogonal in three-dimensional space: (2)$$n_{s}=\vec{C_{ij}p_{1,s}}\times \vec{C_{ij}p_{2,s}},$$
(3)$$N\left(p_{i,j}\right)=\left(\sum_{s\in \left(1,4\right)}\frac{n_{s}}{{∥n_{s}∥}_{2}}\right)/4\;,$$
where *s* denotes different planes ranging from 1 to 4, and $p_{1,s}$ and $p_{2,s}$ are the other two points in the plane apart from the centre pixel point $C_{i,j}$. The surface normal vector perpendicular to the plane $n_{s}$ is obtained by cross-multiplying the vectors of the two points with the centre pixel. Finally, the surface normal vector of the centre point is obtained after normalization and averaging $N\left(p_{i,j}\right)$.

Immediately thereafter, a method for confirming ground points using normalized ground point normals obtained in the camera’s coordinate system is presented. In this system, the right, bottom, and front axes of the camera align with the X, Y, and Z axes in the world coordinate system. An ideal ground normal (denoted as ‘$N={\left(0,1,0\right)}^{T}$’) is chosen as a reference for confirming ground points based on the obtained normalized ground point normals.

The uncertainty in the estimation of surface normals and the non-strict perpendicularity of normalized ground normals are taken into account. The study used the chosen ideal surface normal ($N={\left(0,1,0\right)}^{T}$) with the normal vector of each 3D point in the image for the angle calculation. A threshold close to 0° is set to take into account the possible error introduced by the non-horizontal plane on which the vehicle is located in the image. This threshold controls the identification of “ground point clusters”.

Regarding the calculation of the angular difference, Equation (Equation 4) was chosen to calculate the similarity angle based on the absolute value similarity function of the cosine function:(4)$$S=s\left(P_{i,j}\right)=\left|∠\left(\tilde{n},N\left(P_{i,j}\right)\right)\right|=\left|\arccos\frac{\tilde{n}\cdot P_{i,j}}{∥\tilde{n}∥∥P_{i,j}∥}\right|,$$

After the extraction of ground points from the monocular image and the creation of a dense ground point map, the height from each ground point to the camera is individually estimated. This process yields a substantial dataset of camera heights, and the actual camera height is subsequently determined through a probabilistic calculation method.

The camera height is the projection of the $\vec{OP_{i,j}}$ in the direction of the surface normal at grounding point $N\left(P_{i,j}\right)$, denoting the distance from the camera to the ground point as $\vec{OP_{i,j}}$. The camera height can be calculated using the following equation:(5)$$h\left(P_{i,j}\right)=N{\left(P_{i,j}\right)}^{T}\cdot \vec{OP_{i,j}},$$
where $\vec{OP_{i,j}}$ denotes the vector from the camera to the grounding point.

Due to the large number of ground points, the anomalies hardly affect the estimated value of the true camera height. The obtained median value is determined as the estimated height of the camera, i.e., $h_{M}=Median\left(h\left(P_{i,j}\right)\right)$. In addition to this, the actual height of the camera $h_{R}$ is needed to construct the scale factor as shown in the following equation:(6)$$f_{t}=\frac{h_{R}}{h_{M}},$$

After obtaining the corresponding scale factor, the final absolute depth value is obtained by combining the data obtained from the relative depth map. The following equation shows the method of absolute depth information acquisition by scale factor:(7)$$D_{t}^{abs}=f_{t}\cdot D_{t}^{rel},$$
where $D_{t}^{abs}$ is the calculated absolute distance, $f_{t}$ is the scale factor, and $D_{t}^{rel}$ is the relative distance.

### 2.3. Atmospheric Scattering Model

In foggy conditions, image quality degradation is attributed to the effect of suspended particles in the atmosphere that scatter sunlight and attenuate reflected light. The theory of the atmospheric scattering model proposed by McCartney (McCartney) provides insight into the specific reasons for the loss of colour information and contrast degradation in foggy images. As shown in Figure 3, the foggy day visual imaging model consists of two basic components: an incident light attenuation model and an atmospheric light reflection interference model. The incident light attenuation model describes the energy change in reflected light propagating from the target scene to the acquisition point. Atmospheric light consists mainly of direct light, scattered light from the sky, and reflected light from the ground. As the model shows, the farther the light propagates, the greater the effect of atmospheric light on the fog imaging process. We can see that as the distance between the camera and the object increases, while the effect of atmospheric light increases, the light from the object decreases, leading to a decrease in contrast. The image captured by a typical three-channel RGB (red, green, and blue) camera can be represented by the following equation: (8)$$I_{C}=\int_{\lambda }G_{\lambda }h_{\lambda }R_{\lambda ,C}+V_{\lambda }\left(1-h_{\lambda }\right)R_{\lambda ,C}d\lambda ,$$
(9)$$h_{\lambda }=e^{-d\alpha_{\lambda }},$$
where λ is the wavelength, $V_{\lambda }$ is the airborne light, $h_{\lambda }$ is the transmittance, $\alpha_{\lambda }$ is the fog scattering coefficient, and *d* is the distance between the object and the camera, $C\in \left\{R,G,B\right\}$ is the image channel, and $R_{\lambda ,C}$ is the spectral response of the camera for a particular image channel.

This representation captures the intricate interrelationship between atmospheric light and object light, providing us with a comprehensive understanding of the phenomenon of fog imaging.

### 2.4. Simulated Fog Map for Absolute Depth Estimation

In conditions of dense fog or mist, the scattering coefficient is contingent on the prevailing weather conditions. Specifically, the scattering parameter is derived from the visible range $R_{m}$ through the relationship expressed as $\beta =-\ln\left(\varepsilon \right)/R_{m}$, with its value determined based on an intuitive understanding of the visual range. However, within the visible spectral region, the scattering coefficient $\alpha_{\lambda }$ remains relatively constant. To streamline the algorithm, a constant value is assigned to the wavelength, denoted as $\alpha_{\lambda }=\alpha$ (commonly set to 0.02). Consequently, a simplified atmospheric scattering model is obtained as follows: (10)$$I_{C}\left(x,y\right)=G_{C}\left(x,y\right)h\left(x,y\right)+V_{C}\left(x,y\right)\left(1-h\left(x,y\right)\right),$$
(11)$$h\left(x,y\right)=e^{-\alpha d\left(x,y\right)},$$

In Equation (Equation 10), $\left(x,y\right)$ is the spatial location coordinates of the image, $C\in \left\{R,G,B\right\}$ is the image channel, $I_{C}\left(x,y\right)$ is the image obtained from one of the channels in the image after simulating fog, $G_{C}\left(x,y\right)$ is the fog-free image of a particular channel of the image, and $V\left(x,y\right)$ is the global atmospheric light value, which indicates the influence of other light paths in the atmospheric environment on the direction of perception, which is usually a global constant. In Equation (Equation 11), $h\left(x,y\right)$ is the transmittance, i.e., transmittance map, which is used to describe the ability of light to penetrate the suspended particles in the air on foggy days, and since the scene distances in the three channels of the image can be approximately regarded as the same and the scattering coefficients are almost constant, the transmittance is the same for all the three channels and the value range is roughly from 0 to 1. $V\left(x,y\right)$ is the scattering rate and $d\left(x,y\right)$ is the distance from the detected object to the vision sensor, i.e., the scene depth.

Hence, based on the foggy visual imaging model, the simulation of a foggy scene $I\left(x,y\right)$ can be conceptualized as a process of deriving solutions from an existing fog-free image $G\left(x,y\right)$. Essentially, solving for $I\left(x,y\right)$ entails determining the unknown transmittance $h\left(x,y\right)$ and atmospheric light value $V\left(x,y\right)$ with respect to $G\left(x,y\right)$. In other words, the solution for $I\left(x,y\right)$ involves identifying the unknown transmittance $h\left(x,y\right)$ and atmospheric light value $V\left(x,y\right)$, both of which are contingent on the characteristics of $G\left(x,y\right)$.

Hence, to enhance computational efficiency, we derived a simplified atmospheric scattering model. In the process of obtaining transmittance maps and atmospheric light values for foggy scene simulation, a depth map was acquired using a monocular depth estimation network and dense geometric constraints. Subsequently, Equation (Equation 10) was employed to simulate the dataset image. The subsequent simulation process will be elucidated as shown in Figure 4.

## 3. Fog Day Simulation Image Feature Evaluation Algorithm

### 3.1. Introduction to Evaluation Algorithms

As a key information source for human subjective perception and machine processing systems, images play a decisive role in the accuracy and effectiveness of information acquisition. The quality of images greatly affects the speed and accuracy of subsequent algorithm processing, which highlights the importance of introducing image quality assessment. Considering the subjective nature of human vision, people’s perceptions of fog images appear to be much the same, but quantitative analysis of the features of simulated fog images becomes extremely challenging. In order to verify the authenticity of fog simulated images, an in-depth analysis using a reference-free image quality assessment algorithm facilitates the quantitative assessment of the performance of fog simulated images.

Therefore, in this paper, the simulated fog image realism assessment tool is used to assess the quality of the simulated images in order to obtain the corresponding quantitative assessment indexes. Authenticity Evaluator for Synthetic Foggy/Hazy Images (AuthESI) [34] is a typical natural scene statistics (NSS) feature set based on real fog scenes. It takes the average contrast-normalized luminance coefficient of an image as the basic feature and fits the distribution of mean subtracted contrast normalized (MSCN) coefficients through a generalized Gaussian distribution model. The naturalness of the image is measured using the shape parameters and variance of the fitted Gaussian model. Finally, the quality of the image to be detected is determined using the difference in distance between the parameters of the model characterizing the image to be evaluated and the parameters of the pre-established model. The overall flowchart of AuthESI is shown below:

Figure 5 illustrates a comprehensive flowchart of the selected algorithm, dividing it into two parts: model learning and simulated fog map evaluation. In the model training stage, the distribution of MSCN coefficients for undistorted natural images is approximately Gaussian. Different forms of image distortion will lead to different degrees of distortion in the MSCN coefficient distribution. Taking advantage of this idea, we adopt the generalized Gaussian distribution (GGD) to simulate the distribution of MSCN coefficients. Subsequently, the logarithm of the MSCN coefficients is taken, a step that preserves the correlation between the data attributes and conveniently compresses the variable scales, thus improving the data processing efficiency.

In order to capture the relationship between neighbouring pixels, the logarithmic difference of the MSCN coefficients in the horizontal, vertical, main diagonal, and sub-diagonal directions is derived. Finally, the quality of the evaluated image is determined by measuring the distance between the characteristic model parameters of the target image and the pre-established model parameters.

### 3.2. Comparison of Evaluation Algorithms

In order to accurately verify and evaluate the performance of simulated fog weather, we use four well-known no-reference image quality assessment (NR-IQA) algorithms to experiment and compare with the algorithm in this paper: BRISQUE [35], BLIINDS-II [36], DIIVINE [37], NBIQA [38], and AuthESI [34]. The experiments were executed on the public DHQ database, which employs seven representative dehazing algorithms to generate 1750 dehazed images from 250 real foggy images with varying haze densities. Each image in the database is associated with a corresponding Mean Opinion Score (MOS). Evaluation metrics, including the Pearson Linear Correlation Coefficient (PLCC) and Spearman’s Rank-Ordered Correlation Coefficient (SROCC), were employed to assess the performance of the algorithms. PLCC and SROCC measure the correlation between subjective scores and predicted scores, with a range of [0,1]. Higher values indicate better performance and greater alignment with human visual observations. SROCC primarily assesses monotonicity, while PLCC focuses on prediction accuracy. The experiments utilized a subjective quality database of real foggy images, randomly partitioned into 80–20%, 50–50%, and 20–80% training and test sets. The entire training and testing process was iterated 1000 times, and the average SROCC and PLCC for the five methods were calculated, as illustrated in the Figure 6 below.

As depicted in Figure 6, the AuthESI algorithm emerges as the top-performing model among all considered methods, affirming its efficacy. The figure illustrates that the objective quality scores predicted by AuthESI align more closely with human vision’s perception of image quality in foggy conditions. Through an analysis of the preceding four non-reference image quality evaluation methods, it is evident that, in comparison with AuthESI, their extracted features fall short in capturing the distinctive characteristics of foggy conditions. Consequently, the first four methods are deemed less suitable for assessing the quality of foggy images. Figure 6 demonstrates that the AuthESI algorithm is well-suited for predicting the quality of real foggy images, and its predictions of the authenticity of simulated images exhibit a high degree of consistency with subjective judgments. This assessment tool proves effective in evaluating simulated foggy weather images. In addition, the data show that the algorithm performs relatively better when the training set contains about 50% of the original number of images, suggesting that the algorithm has a strong ability to evaluate foggy images.

AuthESI, the evaluation network for assessing foggy weather consists of two main components: the upper section covers the network training process, and the lower section pertains to the input and detection of simulated fog images. The network training involves utilizing real fog images, and for this purpose, the network is trained on the real fog dataset RTTS. The RTTS dataset, a publicly available real-world foggy day dataset, comprises 4322 real-world foggy images sourced from the Internet, including a substantial collection from traffic and driving cameras. Following dataset screening, 893 images featuring pedestrians, vehicles, and road traffic scenes were selected for training the evaluation network. Upon completing the training, the model’s performance was assessed by calculating the PLCC (Pearson Linear Correlation Coefficient) and SRCC (Spearman’s rank-Ordered Correlation Coefficient) values, which amounted to 0.8124 and 0.8414, respectively. These elevated correlation values signify that AuthESI’s predictions regarding the authenticity of simulated images align more closely with subjective judgments.

## 4. Discussion of Experimental Process

This section provides a concise and precise description of the experimental results, their interpretation, and the experimental conclusions that can be drawn.

### 4.1. Experimental Dataset and Experimental Environment

The BDD100K dataset stands as the most expansive and diverse publicly available autonomous driving dataset to date, released by the Berkeley University AI Laboratory. Comprising 100,000 high-definition videos, each spanning approximately 40 s at 1280 × 720 resolution and 30 frames per second, this dataset serves as a valuable resource for autonomous driving research.

For our study, we selectively sample frames from the 10th second of each video, resulting in 100,000 high-resolution images (1280 × 720) capturing complex traffic scenes. The dataset covers 10 distinct tasks, which effectively amount to 13 classification categories when considering the nuanced distinction of traffic light colours. This multifaceted design aims to facilitate the comprehensive evaluation of the advancements in autonomous driving image recognition algorithms.

Noteworthy for its geographical, environmental, and weather diversity, the BDD100K dataset enables model evaluation across a spectrum of scenarios, fostering robust generalization capabilities. This diverse array of conditions ensures that recognition models are trained to effectively navigate and understand a wide range of real-world driving situations.

### 4.2. Fog Image Simulation Process Analysis

The experimental results are shown below. The image sequence from top to bottom includes the original image, relative depth map, absolute depth map, and the corresponding simulated fog map. The relative depth map uses colours of different brightness to represent different scene depths in the real world. Brighter colours indicate objects that are relatively close to the camera, while darker colours indicate objects that are farther away from the camera. In the absolute depth map, a greyscale scheme is used to represent the absolute depth of the scene, with darker colours indicating objects closer to the camera and lighter colours indicating objects farther away. Subsequently, an algorithm based on the atmospheric scattering model is applied to simulate the corresponding fog scene using the absolute depth information. The results show that the algorithm proposed in this study for simulating fog images achieves excellent simulation results.

As shown in Figure 7b,c, it can be seen from the experimental procedure that this depth estimation method successfully generates a depth map with the same size as the original image. The depth maps contain valuable depth information that accurately reflects the scene in the image, especially at the edges where depth changes are observed. This information is critical for integration with atmospheric scattering models. However, depth map results from monocular depth estimation have some limitations. The edge depth information in Figure 7b relative depth map is imprecise. The absolute depth map Figure 7c, obtained after scale recovery from Figure 7b, shows unexpected depth variations in regions that are expected to be stable. In addition, in some regions of the sky, the scale recovery is imprecise, as shown by the ellipse in Figure 7c, resulting in inaccurate prediction of depth information. These factors reduce the success of the algorithm to a large extent. The evaluation score of the fog map is shown in the upper right corner of Figure 7d, which is intuitively better while scoring lower.

Based on the experimental results, 69,863 clear images in the BDD100K dataset were processed for fog simulation, and 16,702 fog-simulated images were generated with significantly improved results. Although the depth map exhibits some defects, these defects do not explicitly affect the final simulation results of the fog map. However, given the low generation rate, there is a need to improve the depth map algorithm.

### 4.3. Comparison of Simulated Foggy Images under Different Visibility Conditions

Visibility serves as a fundamental meteorological parameter, offering a crucial metric for gauging atmospheric transparency. It signifies the maximum distance at which an observed object can be discerned from its surroundings. The varying levels of visibility correspond to distinct concentrations of fog, as outlined in the Table 1 below.

The fog simulation is performed according to the above relationship between visibility and fog concentration. As shown in Figure 8 below, when the visibility is less than 50 m, it is dense fog; when the visibility is between 50 m and 100 m, it is heavy fog; when the visibility is between 100 m and 200 m, it is moderate fog; and when the visibility is light fog, the visibility is greater than 200 m. They correspond to the following figures.

We select all images in normal scenes as the original input for foggy image generation and process them separately with different foggy densities. To enhance the diversity and complexity of fog density in real fog scenes, we try to simulate fog scenes more realistically and adapt the dataset to multi-density fog scenes. Moreover, it was concluded through experiments that the visibility in foggy days should be limited to a reasonable range, otherwise the artificially generated fog will be almost invisible or too thick to achieve the desired effect.

### 4.4. Night-Time Fog Image Simulation Assessment

Regarding the simulation of nighttime hazy scenes, there are still a number of scholars proposing research methods. Jin et al. [39] proposed a method to accurately simulate nighttime haze by using a 3D scattering formula. Zhang et al. [40] proposed a method to firstly reconstruct the scene geometry then simulate the light and object reflectivity and finally render the haze effect. On the basis of the above, a method to generate realistic night haze images by sampling real-world light colours from a priori distributions was created.

Unlike others, our proposed simulated night fog images are based on self-supervised depth estimation, which is analysed as follows.

As can be observed in Figure 9, in the original nighttime image, nightfall causes the road, the surrounding buildings, and the sky to merge into one, which results in a significant reduction in visibility and a drastic decrease in object recognition. Because of this, there are multiple prediction failures in the generated relative depth map, as shown by the blue boxes in Figure 9. Further observation of the absolute depth map reveals that the objects in the image are almost unrecognisable, which clearly demonstrates the poor generation of the relative depth map, which directly affects the recovery of the image scale and has an impact on the generation of the depth map. The final simulated fog map and its quality score are shown in the upper right corner of the image. It is worth noting that the AuthESI scores of the images are generally high, which indicates that the NSS features of the nighttime fog differ significantly from those of the natural fog. This shows that the algorithm in this paper still has much room for improvement for the simulation of fog images at night.

### 4.5. Evaluation of Foggy Simulated Image Features

In this section, we apply the foggy day simulation framework on the BDD100K dataset and then utilize an image evaluation algorithm to further assess the quality of the foggy images and verify the feasibility of the current foggy image simulation framework. We synthesized a total of 16,702 haze images with improved results from normal scenes in the BDD100K dataset. Given the large amount of data, we employed a Nightingale rose diagram to characterize the quantitative distribution of the data. Originally used to represent seasonal mortality data in military hospitals, the Nightingale rose diagram, also known as a polar plot, resembles the structure of a rose and is adept at conveying information about large amounts of data. The angle of each line segment to the centre of the circle remains constant, while the radius varies depending on the size of the data. In addition, each area can be coloured differently to indicate different data attributes. Essentially, the Nightingale rose chart converts the traditional bar chart into a visually appealing pie chart format. Unlike traditional pie charts, which use angles to represent values or percentages, the Nightingale rose chart uses the radius of a sector to represent the size of the data while using colours to indicate the type of attribute, with each sector maintaining a consistent angle.

We arranged the 16,702 data in descending order, divided them into 60 equal intervals, and then found the percentage of the total number of images in each interval. Finally, we obtained the Nightingale rose diagram shown in Figure 10, which distributes the data from the starting point and completes the combination of the data in a clockwise direction. The graph presents the percentage distribution of images within each scoring interval. The sum of the percentages of data in each interval is 100%. Given the restricted space in the figure and the fact that smaller data had little impact on the analysis of the results, only distributions where the number percentage exceeded 3.5% are presented.

Figure 10 shows the combined scoring results of the algorithm applied to simulated images with a visibility range of 100 m. As can be seen from the Figure 11, the AuthESI values for all images are concentrated in small intervals and the overall values are generally low, with more than 50% of the images scoring below 1.25. This observation suggests that the NSS characteristics of the algorithm are close to those of natural fog.

Evaluation of fog images using the AuthESI metric showed a clear trend towards lower scores overall. Specifically, more than 50% of the evaluated images had scores of no more than 1. This suggests that the scores for the simulated fog images were generally low. In addition, the vast majority (over 90%) of the image scores showed a gradient, indicating a concentration of scores and a relatively low misclassification rate.

After analysing the data with AuthESI scores over 2, we found 1122 images, of which night images accounted for 1001. Combined with the in-depth discussion of the problem in Section 4.4 and the fog map simulation process in Section 4.3, we can comprehensively summarise the reasons for the reduced success rate of the foggy simulation framework proposed in this paper. Firstly, the reduced visibility of the nighttime image leads to inaccuracy in the relative depth estimation, and this inaccuracy has a knock-on effect in the subsequent simulation process, which ultimately affects the quality of the simulated image. Secondly, when processing the sky region of the image, we observed that the relative depth images of the sky region did not have enough contrast to be significant compared to other regions. This lack of contrast has a significant impact in the scale recovery of the absolute depth, which affects the recovery and ultimately leads to the degradation of the fog image quality. Therefore, we believe that the robustness of the relative depth estimation network and the scale recovery network still needs to be further improved to cope with the challenges in complex environments. This consistency in scoring highlights the reliability and stability of the estimation process.

In summary, the above analysis shows that the fog image simulation framework is effective and practical for image simulation under clear weather.

## 5. Conclusions

This study focuses on extracting depth information from monocular clear weather images using a self-supervised monocular depth estimation method and then synthesising foggy weather images based on an atmospheric scattering model. An enhancement algorithm is proposed for generating foggy day image datasets from clear weather conditions.

A depth estimation network is constructed based on a self-supervised monocular depth estimation algorithm to obtain depth maps for the problem of the scarcity of datasets under multi-foggy weather conditions. An atmospheric scattering model is used to estimate transmittance maps from visibility, while dark channel maps are used to distinguish sky regions to estimate atmospheric light values. This approach combines the atmospheric scattering model and the depth maps generated by the depth estimation network to help simulate foggy weather scenarios. The BDD100K dataset was utilised to generate 16,702 (300 m visibility) fog images, which effectively solves the problem of insufficient target detection dataset in foggy weather. As a result, we built a novel foggy weather target detection dataset that can contain three foggy weather scenes with different densities.

In order to verify and assess the authenticity of the synthetic fog images, a non-reference image quality assessment algorithm, AuthESI, is employed. The assessment results show that the algorithm is generally consistent with the features of the natural scene fog data, proving the effectiveness of the proposed algorithm. The practical significance of the algorithm is noteworthy as it introduces a monocular depth estimation network for scene depth estimation without the need for additional sensors, a cost-effective solution for vision-based target detection datasets. Although this study focuses on target detection in foggy weather, it is expected that in future research the method can be extended to tasks such as object recognition and semantic segmentation in other complex scenes, thus broadening its applicability.

## Figures and Tables

**Figure 1 sensors-24-01966-f001:**
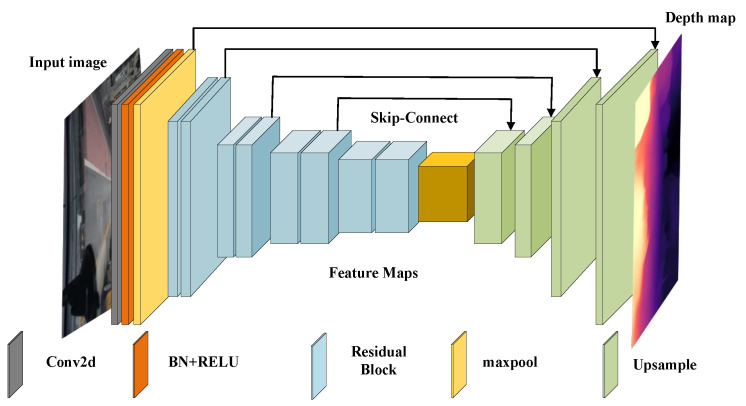
Depth estimation network structure used by Monodepth2.

**Figure 2 sensors-24-01966-f002:**
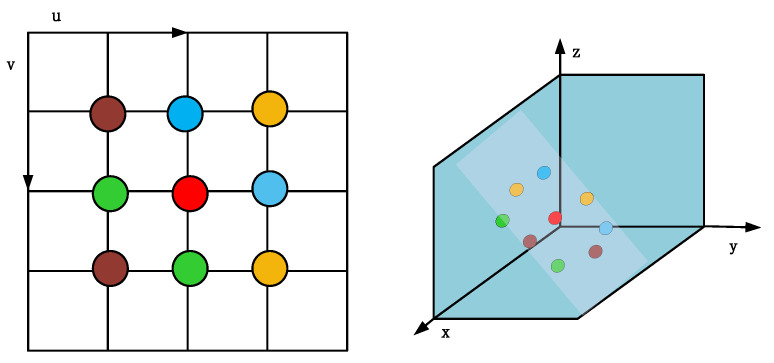
The above figure calculates the projection from 2D space to 3D space and the pairing of 8 neighbouring points. Points of the same colour are paired and then two vectors are formed separately from the centre point, and these two vectors are perpendicular to each other. Four surface normals can be calculated from the four vector pairs and used to form a surface normal.

**Figure 3 sensors-24-01966-f003:**
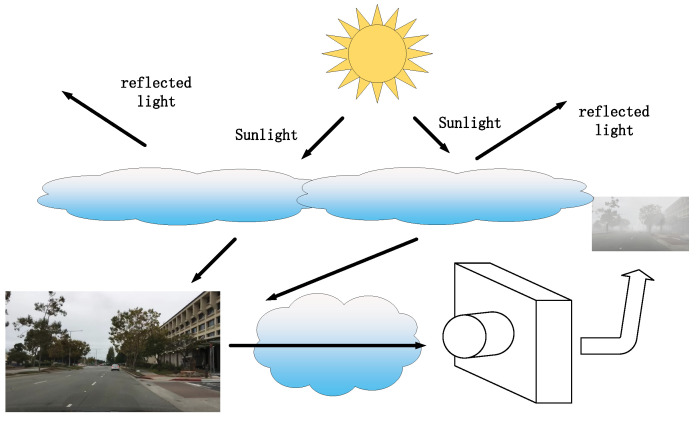
Physical modelling of atmospheric scattering. The light received in the camera sensor consists of light scattered by particles suspended in the air and light reflected by the imaging object, which is attenuated as it travels through the air.

**Figure 4 sensors-24-01966-f004:**
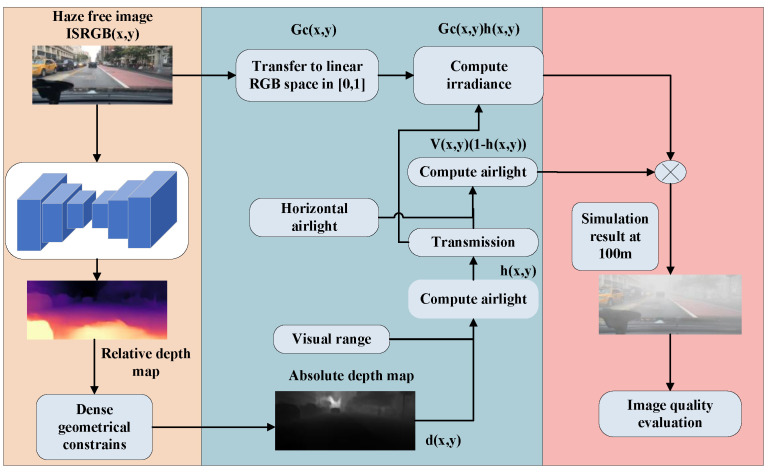
Fog simulation based on Equation (Equation 10).

**Figure 5 sensors-24-01966-f005:**
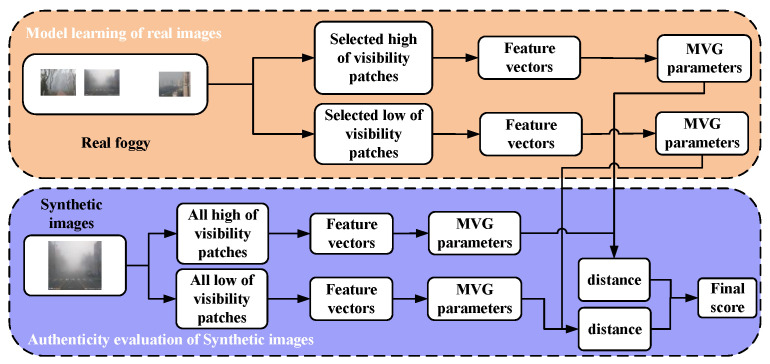
Process flow of AuthESI.

**Figure 6 sensors-24-01966-f006:**
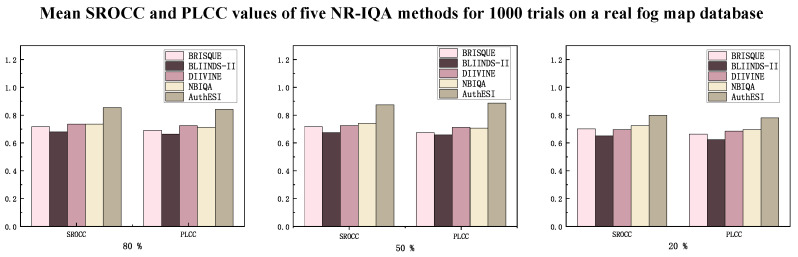
Mean SROCC and PLCC of the five NR-IQA methods across 1000 trials of the DHQ database.

**Figure 7 sensors-24-01966-f007:**
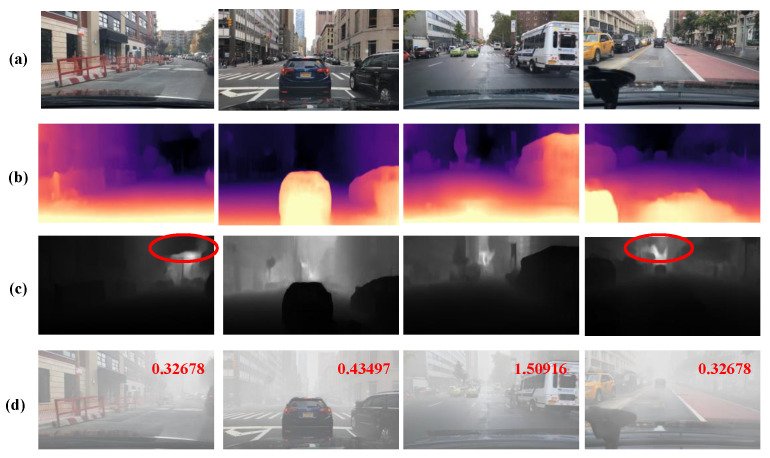
Examples of depth estimation and foggy image synthesis, (**a**) original images from BDD100K, (**b**) relative depth maps predicted by Monodepth2, (**c**) absolute depth maps obtained by dense geometric constraints, and (**d**) synthetic fog (evaluation of the corresponding fog image in the upper right corner of the image).

**Figure 8 sensors-24-01966-f008:**
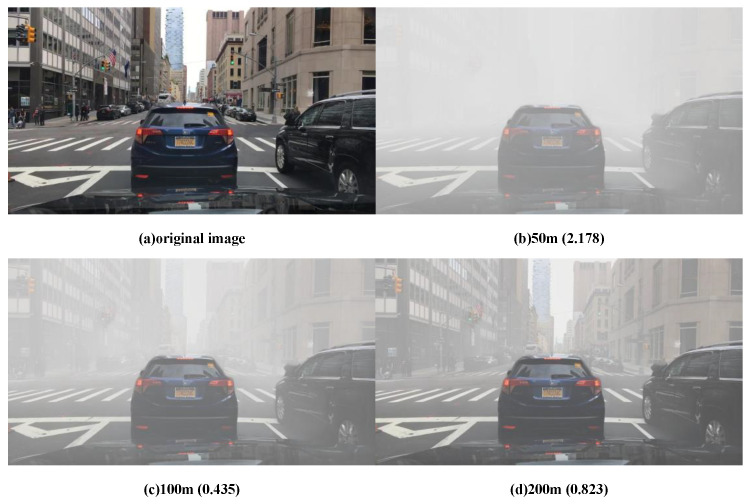
Simulated fog images with scores at three different densities and examples of clear images, with fog thickness decreasing as visibility increases.

**Figure 9 sensors-24-01966-f009:**
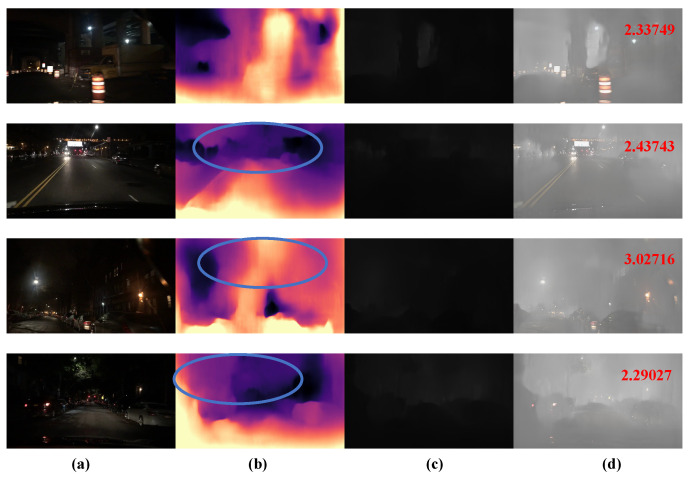
Example of nighttime fog image synthesis, (**a**) original nighttime image at BDD100K, (**b**) relative depth map (the parts circled in blue are significantly less effective relative to the depth map), (**c**) absolute depth map, and (**d**) synthesised fog (quality scores of the corresponding fog images are shown in the upper right corner of the image).

**Figure 10 sensors-24-01966-f010:**
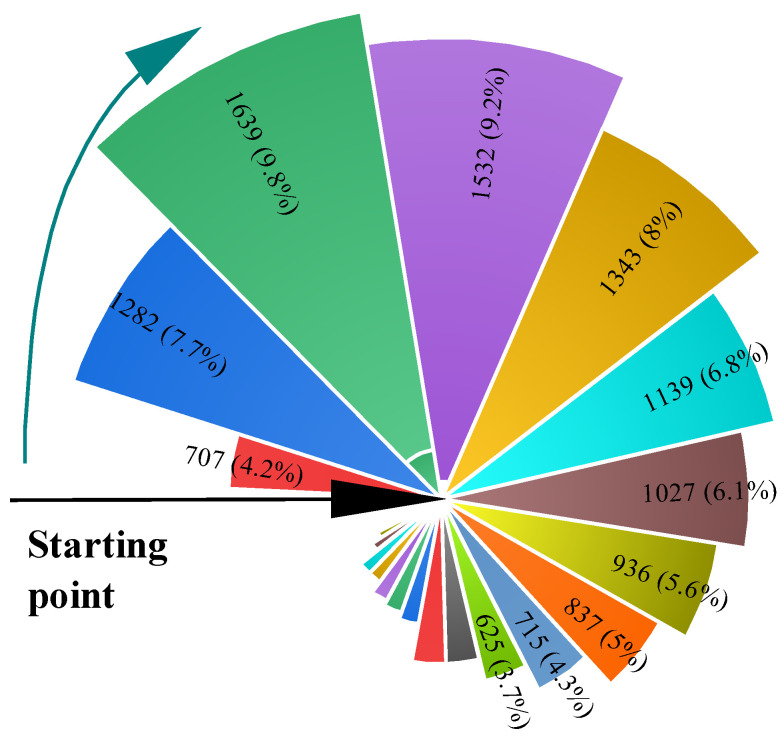
Visibility is the percentage distribution of the number of 100-m simulated fog plots across the different scoring intervals (only cases where the percentage of the number of distributions is greater than 3.5 percent are shown).

**Figure 11 sensors-24-01966-f011:**
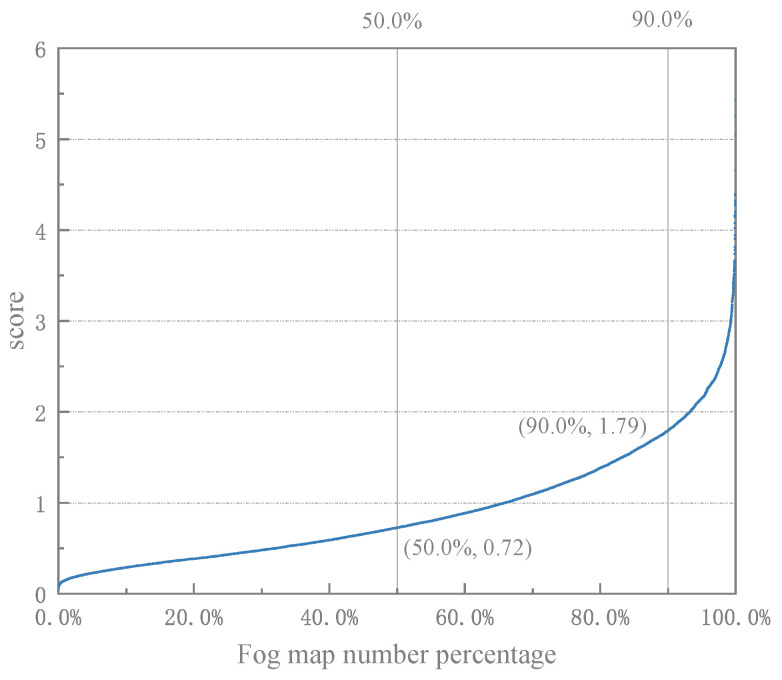
The curve of the data quantity percentage of foggy simulated images changing with the score (visibility 100 m).

**Table 1 sensors-24-01966-t001:** The relationship between road visibility and fog concentration.

Category	Dense Fog	Heavy Fog	Moderate Fog	Light Mist
Visibility/km	<0.05	0.05–0.1	0.1–0.2	>0.2

## Data Availability

Data are contained within the article.

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
