# Peer review of "A Foggy Weather Simulation Algorithm for Traffic Image Synthesis Based on Monocular Depth Estimation"

_sensors, 2024, doi:10.3390/s24061966_

Round 1

Reviewer 1 Report

Comments and Suggestions for Authors

Title: Review Report with Minor Revisions for the Paper Entitled "A Foggy Weather Simulation Algorithm for Traffic Image Synthesis Based on Monocular Depth Estimation"

Abstract: The paper presents a Foggy Weather Simulation Algorithm for Traffic Image Synthesis based on Monocular Depth Estimation to address the challenge of accurate object detection in foggy conditions. The proposed algorithm involves a multi-step process, including self-supervised monocular depth estimation, scale recovery, transmittance map generation, sky region distinction, atmospheric light estimation, and fog simulation using an atmospheric scattering model. Experimental results demonstrate the effectiveness of the proposed method in simulating foggy images closely resembling natural fog, addressing the scarcity of foggy traffic image datasets.

Review:

  1. Citations for AuthESI Scores:
    • The paper could benefit from providing citations for instances where AuthESI scores surpass 2. This would strengthen the evidence of the efficacy and practical utility of the fog image simulation framework.
  2. Literature Review:
    • The literature review should be expanded to include related studies. Notably, the work of ÇODUR M.Y. and KAPLAN N. H. on increasing the visibility of traffic signs in foggy weather (Fresenius Environmental Bulletin, Volume 28, No. 2/2019, pages 705-709) is relevant and can be checked.
  3. Nighttime Discussion:
    • The paper could benefit from a discussion on fog simulation at nighttime, considering the impact on visibility and potential differences in the simulation process. Including relevant citations on nighttime fog simulation would enhance the completeness of the paper.
  4. Keywords:
    • "Traffic safety" should be added to the list of keywords to better reflect the paper's relevance to traffic-related applications.
  5. References Check:
    • The references section should be rechecked to ensure adherence to journal guidelines, particularly regarding dates. Accuracy in referencing is crucial for maintaining the credibility and integrity of the paper.

Conclusion: The paper presents a promising Foggy Weather Simulation Algorithm for Traffic Image Synthesis based on Monocular Depth Estimation. The suggested revisions, including additional citations for AuthESI scores, an expanded literature review, inclusion of nighttime considerations, and adherence to referencing guidelines, will contribute to the overall quality and comprehensiveness of the paper. The proposed fog simulation method has potential implications for traffic safety and addresses the existing limitations in foggy image datasets and incomplete visibility data.

Reviewer 2 Report

Comments and Suggestions for Authors

This is a very significant and important study based on the use of learning-based methods to achieve accurate object detection in foggy conditions. The authors conducts research through an excellent observational study. The paper represents an extremely important segment of research in traffic engineering. The abstract of the paper describes the problem extremely well and indicates the objectives. The authors present an excellent literature review and methodological approach to the research! In the research, specific results of the study were obtained.

Special praise to the authors for the methodology of the paper and presentation of simulation variants. To make the paper even better, I give certain suggestions:

Remove " Monocular Depth Estimation" from the keywords. – It is contained in the title. Add second significant keywords

References 11, 12, 13, 14, 15, 16 and 17 were not used in the paper, and they are listed in the Chapter References

At the end of the introduction, add 2-3 sentences about other chapters in the paper.

Line 57 - Previously, although many algorithms….- State which algorithms they are?

The authors have an excellent conclusion!

Reviewer 3 Report

Comments and Suggestions for Authors

 A research study focused on A Foggy Weather Simulation Algorithm for Traffic Image Synthesis Based on Monocular Depth Estimation

Please provide a thorough discussion on the potential sources of errors in the experimental procedures. A more explicit analysis of systematic ad random errors related to the engine testing and data recording processes is needed.

The basic information regarding the Depth estimation network structure used by Monodepth method must go to the methods not the results section.

Nomenclature is not complete, and the equation parameters must be included.

The conclusion states that the study evaluated the A Foggy Weather Simulation Algorithm for Traffic Image Synthesis with the application of DCP layer explicitly merges features. However, the role and contribution of DCP layer explicitly merges features in the study are not explicitly discussed in the manuscript. Elaborating on the specific applications of Foggy Weather, such as data analysis or model optimization, would enhance the clarity of the methodology and its relevance to the study.

Ensure that the findings are presented in a clear and organized manner. Consider using subsections or bullet points to highlight key outcomes from the study.

Provide quantitative results or statistics to support statements about the improvement in Foggy Weather.

Offer a brief discussion or reference regarding the specific experimental data used for training the foggy weather simulation algorithm. This adds transparency to the methodology

Comments on the Quality of English Language

minor.

Reviewer 4 Report

Comments and Suggestions for Authors

The research addresses the ongoing challenge of learning-based methods to achieve accurate object detection in foggy conditions.

This study primarily investigates the utilization of a self-supervised monocular depth estimation method for extracting depth information from monocular lane images, followed by the synthesis of foggy sky images based on the atmospheric scattering model. An enhanced algorithm for generating foggy image datasets from clear weather conditions is proposed.

The article is well written and I have no reservations to recommending the article for publication.

Author Response

Dear Reviewers,

Thank you very much for reviewing our manuscript, sensors-2897593: A Foggy Weather Simulation Algorithm for Traffic Image Synthesis Based on Monocular Depth Estimation. Once again, we would like to express our deepest gratitude.

Round 2

Reviewer 3 Report

Comments and Suggestions for Authors

Improved quality of Figure 6.

Comments on the Quality of English Language

Minor
